# Tau, Glial Fibrillary Acidic Protein, and Neurofilament Light Chain as Brain Protein Biomarkers in Cerebrospinal Fluid and Blood for Diagnosis of Neurobiological Diseases

**DOI:** 10.3390/ijms25126295

**Published:** 2024-06-07

**Authors:** Yongkyu Park, Nirajan KC, Alysta Paneque, Peter D. Cole

**Affiliations:** 1Rutgers Cancer Institute of New Jersey, 195 Little Albany St, New Brunswick, NJ 08901, USA; nk943@cinj.rutgers.edu (N.K.); amp536@rwjms.rutgers.edu (A.P.); 2Rutgers Robert Wood Johnson Medical School, New Brunswick, NJ 08901, USA

**Keywords:** biomarker, neurological disease, Tau, glial fibrillary acidic protein (GFAP), neurofilament light chain (NfL), cerebrospinal fluid (CSF), blood

## Abstract

Neurological damage is the pathological substrate of permanent disability in various neurodegenerative disorders. Early detection of this damage, including its identification and quantification, is critical to preventing the disease’s progression in the brain. Tau, glial fibrillary acidic protein (GFAP), and neurofilament light chain (NfL), as brain protein biomarkers, have the potential to improve diagnostic accuracy, disease monitoring, prognostic assessment, and treatment efficacy. These biomarkers are released into the cerebrospinal fluid (CSF) and blood proportionally to the degree of neuron and astrocyte damage in different neurological disorders, including stroke, traumatic brain injury, multiple sclerosis, neurodegenerative dementia, and Parkinson’s disease. Here, we review how Tau, GFAP, and NfL biomarkers are detected in CSF and blood as crucial diagnostic tools, as well as the levels of these biomarkers used for differentiating a range of neurological diseases and monitoring disease progression. We also discuss a biosensor approach that allows for the real-time detection of multiple biomarkers in various neurodegenerative diseases. This combined detection system of brain protein biomarkers holds significant promise for developing more specific and accurate clinical tools that can identify the type and stage of human neurological diseases with greater precision.

## 1. Introduction

Biomarkers of neurodegeneration and neuronal injury hold significant potential to improve diagnostic accuracy, disease monitoring, prognosis, and treatment efficacy evaluation. These brain-derived biomarkers are released proportionally into the cerebrospinal fluid (CSF) and blood upon neuronal and astrocytic damage. Tau is known as a member of the microtubule-associated proteins and is a part of the neuronal cytoskeleton. However, in the brain tissue, it is also involved in other vital functions beyond maintaining the cellular architecture [1]. Pathological Tau undergoes aggregation inside neurons, ultimately forming neurofibrillary tangles (NFTs) (Figure 1). The intracellular and extracellular accumulation of different Tau isoforms, including phosphorylated forms, contributes to a diverse group of diseases collectively known as “Tauopathies” [2]. This significant neuropathological and phenotypic heterogeneity within Tauopathies presents a major challenge for developing effective diagnostic and therapeutic strategies.

Glial fibrillary acidic protein (GFAP), another crucial brain protein biomarker, maintains a type-III intermediate filament (IF) network in the astrocytes which represents around 30–40% of cells in the CNS and establish numerous interactions with other cells in the nervous system, including neurons [3]. Astrocytes play a critical role in maintaining synaptic function and supporting axonal metabolism through ion homeostasis regulation (Figure 1) [4]. However, astrocyte injury and degeneration lead to increased GFAP levels in the CSF and blood [5], suggesting their involvement in various neurodegenerative pathologies. Neurofilament proteins (Nfs) are also well suited as biomarkers in these contexts because they are major neuron-specific components that maintain structural integrity and are sensitive to neurodegeneration and neuronal injury across a wide range of neurologic diseases. Nfs belong to the type-IV class of intermediate filaments that are specific to neurons, existing as protein polymers approximately 10 nm in diameter and several micrometers in length [6]. Axonal damage triggers the degradation of Nfs, leading to the accumulation of neurofilament light chain protein (NfL)—a 68 kDa protein abundant in the neurofilament protein complex—within the brain interstitial fluids. Then, these NfL proteins are released into the CSF and blood (Figure 1) [7].

This review summarizes our current understanding of intracellular roles, pathophysiology, and extracellular kinetics of brain protein biomarkers in neurodegenerative diseases. We further explore potential strategies to improve current diagnostic approaches and experimental/clinical trial designs for the more specific and accurate detection of distinct types and stages of human neurological diseases.

## 2. Tau

### 2.1. Tau and Its Structure

Tau, a key member of the microtubule-associated protein family (MAPT), plays a crucial role in maintaining the structural integrity and functional dynamics of neuronal processes [8]. Encoded by the MAPT gene on human chromosome 17q21.3, it exists with six major isoforms in the human brain, varying in their amino-terminal inserts and microtubule-binding repeats [9]. Each isoform contributes to the delicate balance of microtubule stability and flexibility, which are essential for axonal growth, transport, and synaptic function [10]. Tau’s structure consists of four distinct domains: the N-terminal (regulating spacing between microtubules), the proline-rich domain (modulating phosphorylation and binding affinity), the microtubule-binding domain (stabilizing microtubules), and the C-terminal (participating in polymerization) [11]. This intricate architecture enables Tau’s multifaceted interactions with microtubules and actin filaments, shaping the elaborate network of the neuronal cytoskeleton essential for healthy brain function [12].

Alternative splicing of the MAPT gene produces various Tau isoforms, whose relative expression levels change during development, impacting neuronal maturation and axonal growth [9]. Findings suggest that distinct Tau isoforms have specific roles in development, and their imbalanced expression could potentially be involved in the pathogenic processes of neurodegeneration [13]. Importantly, specific Tau isoforms are implicated in various neurodegenerative diseases, highlighting the critical role of Tau structure and function in maintaining neuronal health [2]. Despite being primarily intracellular, Tau can be secreted into the brain’s interstitial fluid, potentially induced by hyperexcitability [14]. In neurodegenerative conditions, the release of Tau into the interstitial fluid is correlated with neuro-axonal damage. The drainage of Tau into the cerebrospinal fluid (CSF) and blood makes the alteration in total Tau levels (t-Tau) a potential indicator of brain pathology, especially in conditions associated with neuronal degeneration. Particularly, increased phospho-Tau (p-Tau) levels in biological fluids reflect ongoing Tau pathology in the brain, representing the hyperphosphorylation of Tau in primary Tauopathies and Alzheimer’s disease (AD) [14].

### 2.2. Tauopathies

Tauopathies represent a diverse group of over twenty neurodegenerative diseases characterized by hyperphosphorylated and aggregated Tau forms in neuronal and glial cells (Table 1) [15,16]. These diseases exhibit varying clinical phenotypes and pathophysiological characteristics [17]. The core feature of Tauopathies is the accumulation of abnormal Tau filaments, forming neurofibrillary tangles (NFTs) and other Tau inclusions in neurons and glial cells (Figure 1) [18]. These aggregations arise from various mechanisms, including abnormal phosphorylation, altered levels, aberrant splicing, or mutations in the Tau gene (MAPT) [19]. Tauopathies exhibit remarkable heterogeneity, which is further classified into primary and secondary forms based on the dominant neurodegenerative agent. Primary Tauopathies such as Pick’s disease and progressive supranuclear palsy are characterized by Tau as the main factor driving neurodegeneration. In contrast, secondary Tauopathies like Alzheimer’s disease exhibit Tau aggregation concurrent with other protein deposits such as beta-amyloid or alpha-synuclein, distinguishing them from primary Tauopathies [20,21]. This interplay between Tau and other amyloidogenic proteins further contributes to the diversity of Tauopathies [18]. Each disease displays distinct clinical presentations, encompassing cognitive decline, behavioral changes, and movement disorders that are influenced by the specific Tau isoforms in the cell types and brain regions [22].

### 2.3. Tau and Its Role in Neurodegenerative Disorders including Alzheimer’s Disease

While a healthy Tau protein plays a critical role in neuronal architecture and stability, its transformation into dysfunctional forms fuels neurodegeneration in Tauopathies. Hyperphosphorylation triggered by kinases such as ERK, JNK, and GSK3β leads to Tau aggregation, forming neurofibrillary tangles (NFTs) that disrupt microtubules by over-stabilizing actin [34]. This cascade leads to neuronal dysfunction, NFT spread, and progressive neurodegeneration. The emergence of diverse animal models and the recognition of independent Tau pathologies, such as those isolated from amyloid-β, challenge the traditional assumption of Tau as solely secondary to amyloid in Alzheimer’s disease [35]. This new understanding emphasizes the crucial role of targeting Tau pathology as a promising therapeutic intervention for combating neurodegenerative disorders. Alzheimer’s disease is a progressive neurodegenerative disorder characterized by the accumulation of abnormal Tau protein and the formation of neurofibrillary tangles (NFTs) [10,36]. The intricate cellular mechanisms underlying Alzheimer’s disease, which Dr. Alois Alzheimer first described the clinical features of in 1906, remained mysterious for decades [37,38]. Recent research has shed light on the pivotal role of Tau protein in disease pathogenesis, revealing a complex interplay between hyperphosphorylation, NFT formation, and neuronal dysfunction. In healthy neurons, Tau proteins act as microtubule-associated proteins (MAPs), essential for stabilizing the intricate network of microtubules that support intracellular transport and cellular architecture. In Alzheimer’s disease, however, Tau undergoes abnormal hyperphosphorylation, altering its structure and disrupting its ability to bind to microtubules [39]. This pathological process leads to the aggregation of Tau into neurofibrillary tangles (NFTs), which accumulate within the neuronal cytoplasm and disrupt vital cellular functions [10].

In addition to Tau, amyloid beta (Aβ) is pivotal in the development of Alzheimer’s disease (AD), marked by the buildup of amyloid plaques in the brain (Figure 1). These plaques primarily consist of Aβ peptides, notably Aβ42 and Aβ40, originating from the amyloid precursor protein (APP). Aβ peptides aggregate into oligomers and fibrils, forming amyloid plaques alongside neurofibrillary tangles made of hyperphosphorylated Tau proteins, defining the neuropathological aspects of AD [3,40,41,42]. The oligomeric Aβ forms are notably neurotoxic, disrupting synaptic function and causing neuronal damage. These Aβ oligomers interfere with cellular processes by compromising membrane integrity, inducing oxidative stress, and hindering synaptic plasticity [42,43,44,45]. Although amyloid plaques are characteristic of AD, interestingly, their presence does not perfectly correlate with disease severity. Instead, the Aβ oligomers closely relate to cognitive deficits in AD patients. Their accumulation leads to neurogenesis impairments and synaptic damage, contributing to cognitive dysfunction [43,45,46,47]. Given Aβ‘s central role in AD, there is a significant interest in measuring Aβ levels in the CSF and plasma as disease biomarkers. The CSF Aβ42 levels, coupled with elevated CSF Tau proteins (t-Tau and p-Tau), are characteristic of AD, facilitating early diagnosis [46,48,49,50]. It turns out that the CSF t-tau/Aβ42 and p-tau/Aβ42 ratios are very robust indicators of AD [24].

### 2.4. Tau as a Biomarker

Tau levels in biological fluids are crucial biomarkers for diagnosing and predicting various neurological conditions. Additionally, analyzing different forms of Tau helps assess disease-modifying therapies and understand molecular mechanisms in neuronal degeneration and Tau pathology. Various techniques, such as ELISA and Western blot analysis, have been used to detect elevated levels of Tau, particularly total Tau (t-Tau), in the CSF of patients with AD (Table 1). In 1993, Vandermeeren established an optimal ELISA technique for CSF t-Tau, demonstrating significantly elevated t-Tau levels in AD patients compared to controls [51]. Following this, three more ELISA techniques were developed in 1995, which are still used today to measure CSF t-Tau. These methods consistently show elevated Tau levels in AD patients [52,53].

#### 2.4.1. CSF Total Tau and Phosphorylated Tau

Elevated t-Tau levels play a decisive role in differential diagnoses, especially in Alzheimer’s disease (AD) where t-Tau is considered a fundamental biomarker, along with phosphorylated Tau (p-Tau) and amyloid beta-42 [53,54]. Total Tau is also a crucial biomarker in various neurodegenerative diseases, such as Creutzfeldt–Jakob disease (CJD) and dementia with Lewy bodies (DLB) [26,55]. Phosphorylated Tau levels in the CSF reflect Tau phosphorylation in the brain with neuronal disease progress, playing an informative role as a biomarker. While total Tau primarily signals axonal degeneration, elevated CSF p-Tau levels are more specific to Alzheimer’s disease [56]. Tauopathies exhibit the abnormal phosphorylation of Tau, leading to the formation of neurofibrillary tangles (NFTs) and impaired axonal transport. Increased CSF concentrations of p-Tau-181, p-Tau-199, and p-Tau-231, as measured by specific assays, help differentiate Alzheimer’s disease from other neurodegenerative diseases (Table 1) [57].

The correlation between phosphorylated Tau, change in neurofibrillary structures, and cognitive decline highlights p-Tau’s crucial role as a diagnostic and prognostic marker in neurodegenerative dementias [58]. The p-Tau/t-Tau ratio is a biomarker measurement in cerebrospinal fluid that helps differentiate conditions such as sporadic Creutzfeldt–Jakob disease (sCJD) from Alzheimer’s disease (AD) and other Tauopathies [59]. This ratio enhances diagnostic accuracy and specificity, providing valuable insights into various neurodegenerative disorders. Other approaches include Tau assays that expand tools to dissect specific Tau isoforms in neurological conditions with altered CSF Tau concentrations [60]. Researchers also developed a novel assay using the 1G2 antibody for a sandwich ELISA, successfully measuring concentrations of non-phosphorylated Tau in human cerebrospinal fluid by specifically targeting epitopes at the 181, 199, or 231 amino acid sites [55].

#### 2.4.2. Plasma Total Tau and Phosphorylated Tau

Blood-based biomarkers present a potential first-line diagnostic tool for Alzheimer’s disease, overcoming limitations of positron emission tomography (PET) scans and CSF analyses. They offer cost-effectiveness compared to PET scans, making them more accessible and applicable globally [61]. Moreover, plasma p-Tau181 also correlates with amyloid and Tau PET, making it a good predictor of brain AD pathology [62]. Initially, measuring Tau in the blood faces challenges due to low assay sensitivity. However, new technological advancements, particularly single-molecule array (Simoa) digital ELISAs, have improved assay sensitivity. Studies using Simoa have found that plasma Tau, especially total Tau, can predict neurodegeneration and may be linked to cognitive decline (Table 1) [1].

Phosphorylated Tau (p-Tau) is present in lower concentrations in blood compared to in the CSF, which makes its detection more challenging. Studies have investigated the potential of plasma phosphorylated Tau species, specifically p-Tau181 and p-Tau231, as biomarkers for Alzheimer’s disease (Table 1). In a sample of 91 subjects, elevated levels of these plasma biomarkers were found in the early stages of AD, comparable to cerebrospinal fluid levels. The study suggests that these plasma biomarkers can serve as effective tools for early AD diagnosis, offering a less-invasive and cost-effective alternative to current methods [63]. Plasma p-Tau, specifically p-Tau181, is also highlighted as a potentially reliable alternative to CSF p-Tau181. Plasma p-Tau offers a less-invasive, timesaving, cost-effective, and easily accessible option for detecting early Tau deposition in future clinical practice and trials [64,65].

## 3. GFAP

### 3.1. Basic Information about GFAP

Glial fibrillary acidic protein (GFAP) is a main intermediate filament protein (IF) within astrocytes, forming a network that maintains their structure and resilience. GFAP, initially identified as a marker for multiple sclerosis [66], is described as a single acidic protein that is the major component of astrocytes, and it differs from other fibrillary proteins in structure and expression [66]. Since this publication, more research has been published establishing GFAP as an essential protein in the cytoskeleton of glial cells, responsible for neuronal mechanical strength and the maintenance of the blood–brain barrier [67]. GFAP assembly is regulated by various post-translational modifications, including phosphorylation, dephosphorylation, glycosylation, and citrullination [68,69]. The roles of these modifications in GFAP assembly are still rather unclear. However, some studies have proposed that the specific tail sequences of GFAP determine GFAP’s interactions with other intermediate filaments (IFs) and proteins [68,69]. Clinically, GFAP serves as a biomarker for astrogliosis, with elevated levels observed in brain injuries and neurodegenerative diseases (Table 1) [69].

### 3.2. GFAP and Its Role in Neurodegenerative Diseases

GFAPs increase as a byproduct of astrogliosis (astrocytosis) from a stress response of the CNS where astrocytes proliferate dramatically due to cell death and the degeneration of surrounding neurons [67]. Evidence suggests that during astrogliosis, GFAPs and breakdown products are rapidly released into the CSF and blood [67]. Thus, elevated GFAP, detected in CSF and blood, is present in numerous neurodegenerative pathologies due to the breakdown of the astrocyte cytoskeleton (Figure 1). Studies have evaluated various levels of GFAP increase in several pathologies such as Alzheimer’s disease (AD), Lewy body dementia (DLB), frontotemporal lobar degeneration (FTLD), Creutzfeld–Jakob disease (CJD), and Parkinson’s disease (PD) (Table 1) [5,27,70]. A significant elevation in CSF GFAP was found in patients with AD, DLB, and FTLD when compared to healthy controls, presumably due to the degeneration of astrocytes, releasing GFAP as a byproduct [5]. Among these neurodegenerative pathologies, GFAP levels in FTLD patients were significantly higher compared to those in AD and DLB. These findings may be attributed to the level of involvement of astrocytes and gliosis in various disease processes. In FTLD, astrogliosis is the primary mechanism of brain injury, and blood GFAP levels are associated with GABAergic deficits. In other words, an increase in blood GFAP may be coupled with heightened impulsivity and decision-making deficits, indicating a more severe clinical progression of FTLD [71,72]. While astrogliosis is present also in AD, it is less than in FTLD. AD is predominantly characterized by the involvement of amyloid plaques and Tau protein, and to a lesser extent astrogliosis [73].

GFAP on its own is not indicative of any specific neurodegenerative disease, but it is elevated in the CSF and blood of most neurodegenerative diseases when compared to control groups. One study comparing the blood GFAP levels in AD, FTLD, and other neurodegenerative dementias found that the GFAP level can differentiate between AD and FTLD neurodegenerative processes from healthy controls [29]. However, this study indicated that aside from AD and FTLD, other blood biomarkers such as Tau and NfL are more accurate in differentiating among neurodegenerative diseases [29]. Many neurodegenerative processes often overlap or present as mixed pathologies, which complicates diagnosis. In cases of overlapping pathologies, the GFAP level can be used to confirm the presence of neurodegenerative processes, but other markers may be more accurate in distinguishing between the diagnoses.

### 3.3. Elevated GFAP Levels Following Traumatic Brain Injury (TBI)

Recovery from traumatic brain injury (TBI) is limited by early diagnosis and treatment. While traditional methods like CT scans and GCS are used, they have limitations, such as negative CTs for mild TBI and sedation affecting GCS evaluation [74,75]. In addition, each exposure to the radiation required for CT scanning incrementally increases the risk of radiation-related malignancy [76]. This has led to the exploration of biomarkers, and in 2018, the FDA (US Food and Drug Administration*) approved a blood test measuring GFAP and other proteins (e.g., UCH-L1) for evaluating mild TBI. For use of this test, blood protein levels are measured within 12 h of injury with a sensitivity of 97.5% and a specificity of 99.6%. GFAP levels peak within 1 h of initial injury and deplete within 24–72 h, so measuring GFAP levels in the serum is immediately necessary. One observational study showed that GFAP levels measured within 12 h of TBI have a prognostic value for death and poor outcome but have little value in predicting recovery by the 6-month mark [77]. Several studies have established GFAP cutoff values for TBI diagnosis (Table 1). One study involving 1900 mild-to-moderate TBI patients identified a combination of 22 pg/mL serum GFAP and 327 pg/mL UCH-L1 as predictive of intracranial injury [78]. Another study identified patients with traumatic brain injury with a GFAP cutoff value of 230 pg/mL [30]. Consistently, across numerous studies, GFAP demonstrates the highest predictive and diagnostic value for TBI compared to other biomarkers [30,78].

*: The FDA authorizes the marketing of first blood tests to aid in the evaluation of concussion in adults. https://www.fda.gov/news-events/press-announcements/fda-authorizes-marketing-first-blood-test-aid-evaluation-concussion-adults (accessed on 13 February 2018).

### 3.4. GFAP and Brain Cancers

GFAP serves as a biomarker for tumor progression and treatment response in various brain cancers, including astrocytomas, glioblastomas, ependymomas, and metastases [79,80,81,82]. GFAP is a well-known marker of astrocytoma. Elevated CSF GFAP levels were originally believed to be an indicator of a less-malignant, more-differentiated astrocytoma, but recent literature has suggested that distinguishing between individual GFAP isoforms can improve the outcomes of treatment. Some studies suggest that two isoforms of GFAP, GFAPα and GFAPδ, must be compared to make a true judgment on malignancy. GFAPδ has a higher malignant behavior when compared to GFAPα, proposing that different GFAP isoforms behave differently within cancer cells [83]. Interestingly, cancer can trigger an autoimmune response against GFAP, leading to neurological symptoms. A case study reported that a 76-year-old woman with breast cancer was also diagnosed with GFAP astrocytopathy (an autoimmune CNS disorder), which is triggered by anti-GFAP antibodies, likely secondary to breast cancer. Notably, her GFAP astrocytopathy was improved with breast cancer therapy, suggesting a potential paraneoplastic neurological syndrome [84].

### 3.5. Conclusions and Limitations of GFAP

Compelling evidence demonstrates that GFAP can and should be used as a biomarker for brain injury across various disease processes. The extent to which GFAP rises in the CSF and blood depends on the pathology causing the CNS injury. For instance, while the half-life of GFAP is typically 24–72 h, it can remain elevated for weeks in certain inflammatory conditions. In such cases, distinguishing between ongoing astrocyte damage and the clearance of previously damaged cells over time becomes challenging [85]. While GFAP can sometimes offer valuable prognostic insights, information about specific isoforms like GFAPα and GFAPδ is often crucial. For example, these isoforms hold distinct significance in assessing the malignancy and differentiation of cancer cells in astrocytoma [83].

## 4. NfL

### 4.1. NfL Structure and Function

Neurofilaments (Nfs) are cylindrical proteins primarily located in the neuronal cytoplasm (Figure 1). They consist of five major Nf subunits: neurofilament light chain (NfL), neurofilament medium chain (NfM), neurofilament heavy chain (NfH), alpha-internexin (INA) and/or peripherin (PRPH) [86]. Each Nf subunit includes an amino-terminal globular head domain, a conserved alpha-helical rod domain, and a carboxy-terminal tail domain. Notably, the variable amino-terminal head and carboxy-terminal tail domains contain numerous phosphorylation and O-linked glycosylation sites [32]. Studies in mice lacking the C-terminal region of Nf subunits (which contains most phosphorylation sites) suggest that Nf phosphorylation regulates axonal Nf dynamics, including axonal transport rates [87]. C-terminal phosphorylation of Nf subunits is regionally/temporally regulated by a balance of kinase and phosphatase activities. Misregulation of this balance contributes to motor neuron diseases. Nf monomers assemble into coiled-coil heterodimers, then tetramers, and subsequently into unit-length filaments. These filaments undergo end-to-end longitudinal annealing, leading to the formation of mature Nfs with a diameter of 10 nm after radial compaction [32]. Nfs are classified as intermediate filaments (IFs) due to their diameter (10 nm), which falls between actin (6 nm) and myosin (15 nm). Nfs provide structural stability to neurons and are present in dendrites, neuronal soma, and axons, where their expression is particularly high. Since Nfs enable the radial growth of axons, larger myelinated axons abundantly express Nfs [6]. Among the three neurofilament chains, NfL is the most abundant and soluble subunit in the Nf complex.

### 4.2. NfL Release to Brain Interstitial Fluid

The identification and quantification of axonal damage are indicators that allow for diagnostic accuracy and prognostic assessment in the management of neurological diseases. Because neurofilament light chain (NfL) is a highly expressed neuronal cytoplasmic protein in the myelinated axons, the axonal damage and degeneration release NfL to the brain interstitial fluid (Figure 1). Under physiological conditions, low levels of NfL are constantly released from neurons into the extracellular space throughout normal brain development, maturation, and aging. This released NfL freely distributes to the CSF and eventually reaches the blood, where its concentration is roughly 40-fold lower than in the CSF (Figure 1) [88]. Under normal conditions, NfL release from axons exhibits an age-dependent pattern, with higher levels observed in older individuals [86]. In a study of healthy individuals aged 31–56, log CSF NfL showed a strong linear correlation with age (r = 0.77, *p* < 0.0001), presenting a yearly increase of 3.1% in CSF NfL [89]. However, in various neurological disorders, this release of NfL significantly increases in the CSF and blood proportionally to the degree of axonal damage. This includes inflammatory, neurodegenerative, traumatic, and cerebrovascular diseases. Regardless of the cause, higher NfL levels in the CSF and blood are correlated in response to neuronal injury and neurodegeneration (Table 1). Among Nf subunits, the phosphorylated neurofilament heavy chain (pNfH) influences the dynamics of Nf transport along axons and axonal stability [6]. Therefore, elevated pNfH in the CSF may act as a biomarker of axonal injury, especially in amyotrophic lateral sclerosis (ALS), where it shows high specificity [90]. However, NfL remains the most reliably measurable Nf subunit in biofluids due to its abundance and solubility as the Nf backbone.

### 4.3. NfL Biomarker in Neurological Disorders

While blood NfL levels are lower than NfL concentrations in the cerebrospinal fluid (CSF), a strong correlation of NfL exists between the two biofluids. Therefore, blood NfL levels also serve as a marker of CNS neuronal degeneration. However, the NfL is an unspecific marker of neuronal degeneration, meaning that it cannot pinpoint the underlying cause of the degeneration [86,91]. Nonetheless, the level of NfL is extremely useful in monitoring the progression of demyelinating diseases [32,86]. Therefore, precisely monitoring NfL in the CSF and blood has the potential to discover important clues for the development of therapeutic and diagnostic approaches for various psychiatric diseases and neurodegenerative diseases such as Parkinson’s disease and Alzheimer’s disease [92,93]. It is widely acknowledged that the pathophysiology underlying many neurodegenerative disorders, such as Alzheimer’s disease (AD), originates many years prior to clinical symptoms. AD progresses through three stages—an early, preclinical stage with no detectable symptoms; a middle stage of mild cognitive impairment; and a late stage marked by symptoms of dementia. There is a growing need for reliable non-invasive blood-based biomarkers for AD that can facilitate diagnosis, predict disease progression, and provide evidence of disease modification. NfL is rapidly emerging as a transformative blood biomarker in neurology, providing novel insights into a wide range of neurological diseases and advancing clinical trials. In both familial and sporadic Alzheimer’s diseases, blood NfL levels are observed to rise as early as 22 years before familial AD clinical appearance and 10 years before sporadic AD [32,94]. Although amyloid-beta and Tau proteins are widely regarded as useful diagnostic biomarkers of AD, the Tau proteins are unaltered in other neurodegenerative diseases, such as Tau-negative frontotemporal dementia (FTD), caused by granulin or C9orf72 mutations. In contrast, NfL fragment levels in such patients are eight times higher compared to healthy controls [95]. Furthermore, in Huntington’s disease (HD), CSF NfL fragment levels exhibit a stronger correlation with disease progression than CSF Tau levels [96]. This suggests that NfL and its fragments, as general markers of neuronal integrity, may be more sensitive to neurodegeneration than Tau.

In ischemic stroke, neurofilament network damage contributes to the transition toward long-term tissue damage. When ischemia-affected tissues were analyzed with a multi-parametric approach including immunofluorescence labeling, Western blotting, and electron microscopy, the impaired NfL proteins were shown to be degraded in these ischemic stroke-damaged tissues from three stroke animal models of middle cerebral artery occlusion (MCAO), as well as human autoptic stroke tissue [97]. Further, multiple fluorescence labeling of neurofilament proteins revealed the spheroid and bead-like structural alterations in human and rodent tissue, correlating with cellular edema and cytoskeletal disorder at the ultrastructural level [97]. These consistent findings across animal and human tissues suggest that neurofilaments are promising cytoskeleton backbones for neuroprotective strategies aimed at maintaining their integrity. To investigate neurofilament light (NfL) as a CNS biomarker in patients with systemic lupus erythematosus (SLE) and primary Sjögren’s syndrome (pSS), the NfL levels in the CSF were measured by exploring associations with clinical, structural, immunological, and biochemical abnormalities. In SLE patients, higher NfL concentrations were associated with impairments in psychomotor speed and motor function, and in pSS with motor dysfunction [98]. The role of NfL as a biomarker has been largely reported in a variety of neurological diseases, including multiple sclerosis (MS), Alzheimer’s disease (AD), frontotemporal dementia (FTD), amyotrophic lateral sclerosis (ALS), atypical parkinsonian disorders (APD) and traumatic brain injury (TBI) (Figure 2). CSF NfL levels are consistently higher in patients with neurological diseases compared to healthy controls (HC) and similar findings have been reported also with blood NfL levels [86]. Particularly in Creutzfeldt–Jakob disease and neurological complications of HIV infection, NfL levels reach very high concentrations in the CSF (over 30-fold higher than in healthy conditions, as shown in Figure 2).

## 5. Discussion of Biomarker Detections

Given the distinct roles among brain protein biomarkers Tau, GFAP, and NfL, the growing interest in the precise and quantitative detection of these biomarkers underscores their potential for diagnosing neurodegenerative diseases. For the detection of these biomarkers, the enzyme-linked immunosorbent assay (ELISA) stands out as a convincing tool that is frequently used to quantify proteins, antibodies, antigens, and glycoproteins in biological samples. Examples include pregnancy tests, diagnosing HIV infection, and quantifying cytokines or soluble receptors in cell supernatant or serum [99]. In the CSF, brain protein biomarkers can be measured using sandwich ELISA technology [100,101]. However, the sensitivity of ELISA for measuring blood biomarker concentration is not sufficient [102]. New ultra-sensitive methods now enable the minimally invasive measurement of these low levels of biomarkers in serum or plasma, enabling the tracking of disease onset and progression in neurological disorders or nervous system injuries and assessing responses to therapeutic interventions. Biosensor platforms represent another exciting advancement, utilizing bioreceptors and transducers to convert biological interactions into measurable electrical signals. These platforms hold immense potential for real-time, multi-biomarker detection in clinical settings [103,104]. The biological elements such as enzymes, antibodies, and nucleic acids in the bioreceptor interact with the analytes under test, and the transducer converts the biological response into an electrical signal [105]. Three major transducer-based groups have been developed over the last two decades, allowing different bioreceptors to be applied to different biological targets [100]. Biosensors, such as nano-bioelectronics, are crucial tools to advance the detection of new specific biomarkers that are molecularly relevant to specific types and stages of human diseases [106,107]. Furthermore, nanotechnology plays a crucial role in developing highly sensitive and minimally invasive procedures for detecting trace amounts of biomarkers, particularly in the early stages of neurodegenerative diseases. This paves the way for earlier diagnosis and intervention, potentially improving patient outcomes. Nonetheless, the use of these biomarkers in clinical practice is still evolving, and further research is needed to establish their utility in routine diagnostic testing.

## 6. Conclusions

The neurofilament light chain (NfL) proteins are released from neurons into the extracellular space proportional to the degree of axonal damage in a variety of neurological disorders (Figure 1). Since the NfL level is a sensitive but unspecific marker of axonal injury cause, its potential diagnostic value does not lie in the ability to differentiate between neurological diseases with similar degrees of axonal loss. For these reasons, the potential diagnostic role of NfL proteins in the clinical setting should be complemented with other neurological assessments, as well as more disease-specific biomarkers and brain imaging findings. Glial fibrillary acidic protein (GFAP) maintains the intermediate filament (IF) network in the astrocyte. Increased GFAP levels in the cerebrospinal fluid often indicate astrocyte degeneration through processes like astrogliosis. While Tau/p-Tau proteins are well established as diagnostic biomarkers for Alzheimer’s disease, their presence can also be detected in other neurodegenerative conditions. With the biosensor approach, which allows for the real-time detection of multiple biomarkers in various neurodegenerative diseases, a combined approach of brain protein biomarkers—NfL, GFAP, and Tau—holds significant promise for developing more specific and accurate clinical tools that can identify the type and stage of human neurological diseases with greater precision.

## Figures and Tables

**Figure 1 ijms-25-06295-f001:**
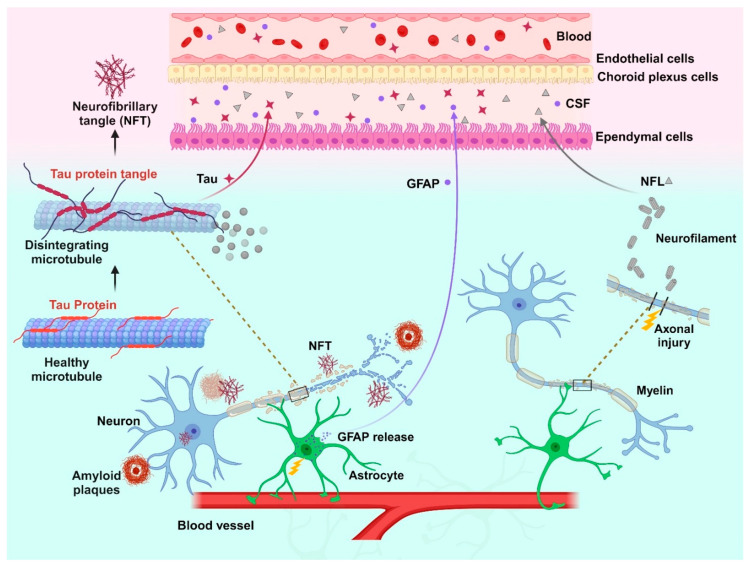
Schematic representation of neurodegenerative disease biomarkers in CSF and blood. This figure illustrates the release of three important brain protein biomarkers in neurodegenerative diseases: Tau, glial fibrillary acidic protein (GFAP), and neurofilament light chain (NFL) protein. In healthy individuals, NfL is a structural component of the neuronal cytoskeleton, while Tau protein stabilizes microtubules. GFAP forms part of the intermediate filaments within astrocytes. Upon neuronal injury or death, NfL is released into the interstitial fluid and subsequently enters the cerebrospinal fluid (CSF) and blood. Similarly, in cases of astrocyte damage or activation, GFAP is released into the extracellular space and can be detected in the CSF and blood. Tau protein forms intracellular aggregates in diseased neurons and can also be released into the CSF and blood, to a lesser extent.

**Figure 2 ijms-25-06295-f002:**
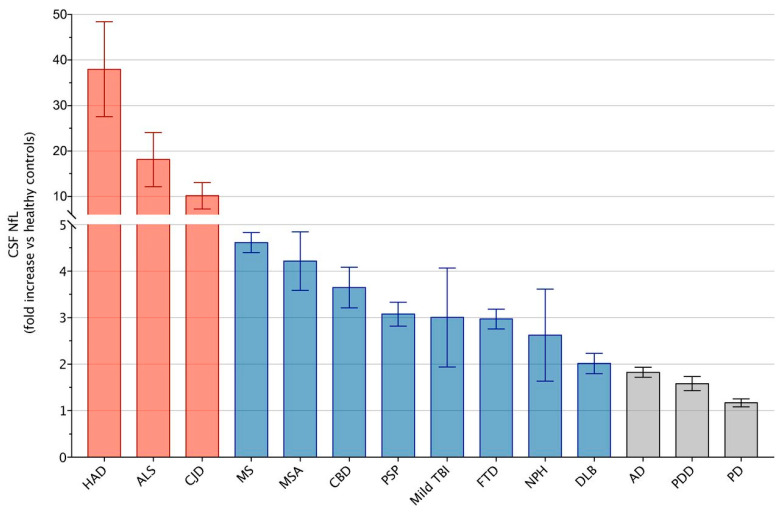
Fold increase of neurofilament light chain (NfL) in cerebrospinal fluid (CSF) of neurological diseases associated with axonal damage. Columns represent mean fold increases and SEM of CSF NfL in neurological diseases versus healthy controls (HC). HAD: HIV-associated dementia; ALS: amyotrophic lateral sclerosis; CJD: Creutzfeldt–Jakob disease; MS: multiple sclerosis; MSA: multiple system atrophy; CBD: corticobasal degeneration; PSP: progressive supranuclear palsy; Mild TBI: mild traumatic brain injury; FTD: frontotemporal dementia; NPH: normal pressure hydrocephalus; DLB: dementia with Lewy bodies; AD: Alzheimer’s disease; PDD: Parkinson’s disease dementia; PD: Parkinson’s disease. Adapted from [86].

**Table 1 ijms-25-06295-t001:** Physiological concentrations of three biomarkers Tau, GFAP, and NfL in Blood and CSF. Some physiological concentrations are indicated multiple times with separation (;), which are cited from several different references.

	Disease	Blood (pg/mL)	CSF (pg/mL)
t-Tau	Normal	1.8 ± 0.5 [23]	213 [24]
	Alzheimer’s disease (AD)	2.5 ± 1.3 [23]	318 [24]; 601 [25]; 604 [26]
	Parkinson’s disease dementia (PDD)	3.1 [27]	
	Creutzfeldt–Jakob disease (CJD)	9.0 [28]	2060 [26]; 6520 [28]
	Frontotemporal lobar degeneration (FTLD)		350 [26]
	Dementia with Lewy bodies (DLB)		305 [26]
	Vascular dementia (VaD)		238 [26]
	Corticobasal degeneration (CBD)		262 [26]
	Subjective memory complaints (SMC)		245 [26]
	Mild cognitive impairment (MCI)		246 [24]; 310 [25]
p-Tau	Normal		18 [24]
	Mild cognitive impairment (MCI)		22 [24]; 53 [25]
	Alzheimer’s disease (AD)		29 [24]; 78 [25]; 83 [26]
	Frontotemporal lobar degeneration (FTLD)		47 [26]
	Dementia with Lewy bodies (DLB)		52 [26]
	Vascular dementia (VaD)		35 [26]
	Corticobasal degeneration (CBD)		50 [26]
	Progressive supranuclear palsy (PSP)		36 [26]
	Creutzfeldt–Jakob disease (CJD)		54 [26]; 61 [28]
	Psychiatric disorder (PSY)		41 [26]
	Subjective memory complaints (SMC)		45 [26]
	p-Tau181: Normal	2.6 ± 1.0 [23]	
	p-Tau181: Alzheimer’s disease (AD)	5.6 ± 2.0 [23]	>65.5 (cutoff) [29]
	p-Tau231: Normal	7.5 ± 1.6 [23]	
	p-Tau231: Alzheimer’s disease (AD)	14.6 ± 6.1 [23]	
GFAP	Normal	79.0 [25]	2175 [5]
	Alzheimer’s disease (AD)	404.7 [29]	2990 [5]
	Frontotemporal lobar dementia (FTLD)	198.2 [29]	4780 [5]
	Mild cognitive impairment (MCI)	167.5 [25]	
	Traumatic brain injury (TBI)	>230.0 (cutoff) [30]	
	Parkinson’s disease dementia (PDD)	145.8 [27]	
	Creutzfeld–Jakob disease (CJD)	815.0 [28]	
	Relapsing remitting (+) multiple Sclerosis (MS)	129.8 [31]	
	Lewy body dementia (DLB)		3400 [5]
NfL	Normal	8.1 [25]	284.4 [32]; 584.1 [25]
	Mild cognitive impairment (MCI)	12.9 [25]	807.7 [25]
	Alzheimer’s disease (AD)	15.5 [25]; 21.9 [29]	1559.0 [25]
	Frontotemporal lobar dementia (FTLD)	44.9 [29]	>1801 (cutoff) [29]
	Creutzfeld–Jakob disease (CJD)	116 [28]	7500 [28]
	Parkinson’s disease dementia (PDD)	21.6 [27]	
	Relapsing remitting (+) multiple sclerosis (MS)		951.8 [32]; >807.5 (cutoff) after 1 year increases risk of relapse ×5 [33]

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
