# Peer review of "Tau, Glial Fibrillary Acidic Protein, and Neurofilament Light Chain as Brain Protein Biomarkers in Cerebrospinal Fluid and Blood for Diagnosis of Neurobiological Diseases"

_ijms, 2024, doi:10.3390/ijms25126295_

Round 1
Reviewer 1 Report (Previous Reviewer 1)
Comments and Suggestions for Authors
No further comments.
Reviewer 2 Report (Previous Reviewer 3)
Comments and Suggestions for Authors
The article has been modified as per the comments and the Figures and legends may be rearranged and shortened
This manuscript is a resubmission of an earlier submission. The following is a list of the peer review reports and author responses from that submission.
Round 1
Reviewer 1 Report
Comments and Suggestions for Authors
In this review manuscript the authors summarized importance of brain protein biomarkers such as Tau, Gfap and Nfl for diagnosis of neurobiological diseases. Although this is an interesting subject, there are several comments that should be clarified further by the authors:
Comments:
1. Important information with regard physiological concentrations of three biomarkers Tau, Gfap and Nfl in CSF and plasma are missing in the manuscript. The authors must include the published numbers in the text or in form of a table.
2. The conclusion (pages 13 and 14) is too long. This should be shortened in one paragraph.
3. Figure1: What are types of cells in red, yellow and green? These cells should be labeled with names.
4. Figure 1: Amyloid plaques are shown in Figure 1. However, there is no information in the manuscript whether beta-amyloid levels in plasma and CSF can be used as biomarkers for the diagnosis of neurobiological diseases as well. The authors need to address this issue.
Reviewer 2 Report
Comments and Suggestions for Authors
The title of the paper refers to The measurement of these biomarkers in CSF and blood is an active area of research, and their potential as diagnostic and prognostic tools for neurobiological diseases is being investigated. However, it's important to note that the use of these biomarkers in clinical practice is still evolving, and further research is needed to establish their utility in routine diagnostic testing. The article took into account the research in the field, the methodology reflects the work of the authors in the field and the results and conclusions allow the study to be accepted for publication. In this paper, the DISCUSSIONS chapter is missing.
Reviewer 3 Report
Comments and Suggestions for Authors
The information presented are mostly available and not comprehensively compiled and hence this will not add much to the scientific world

The information presented are mostly available and not comprehensively compiled and hence this will not add much to the scientific world